METHODS

# Implicit model to capture electrostatic features of membrane environment

**Rituparna Samanta**[1], **Jeffrey J. Gray**[1,2,3]*

**1** Department of Chemical and Biomolecular Engineering, Johns Hopkins University, Baltimore, Maryland, United States of America, **2** Program in Molecular Biophysics, Johns Hopkins University, Baltimore, Maryland, United States of America, **3** Sidney Kimmel Comprehensive Cancer Center, Johns Hopkins School of Medicine, Baltimore, Maryland, United States of America

* jgray@jhu.edu

## Abstract

Membrane protein structure prediction and design are challenging due to the complexity of capturing the interactions in the lipid layer, such as those arising from electrostatics. Accurately capturing electrostatic energies in the low-dielectric membrane often requires expensive Poisson-Boltzmann calculations that are not scalable for membrane protein structure prediction and design. In this work, we have developed a fast-to-compute implicit energy function that considers the realistic characteristics of different lipid bilayers, making design calculations tractable. This method captures the impact of the lipid head group using a mean-field-based approach and uses a depth-dependent dielectric constant to characterize the membrane environment. This energy function Franklin2023 (F23) is built upon Franklin2019 (F19), which is based on experimentally derived hydrophobicity scales in the membrane bilayer. We evaluated the performance of F23 on five different tests probing (1) protein orientation in the bilayer, (2) stability, and (3) sequence recovery. Relative to F19, F23 has improved the calculation of the tilt angle of membrane proteins for 90% of WALP peptides, 15% of TM-peptides, and 25% of the adsorbed peptides. The performances for stability and design tests were equivalent for F19 and F23. The speed and calibration of the implicit model will help F23 access biophysical phenomena at long time and length scales and accelerate the membrane protein design pipeline.

**Data Availability Statement:** The F23 energy function is available in the Rosetta software suite (https://www.rosettacommons.org). Rosetta is available to noncommercial users for free and to commercial users for a fee. The scripts used to run

## Author summary

Membrane proteins participate in many life processes. They constitute 30% of the human proteome and are targets for over 60% pharmaceuticals. Accurate and accessible computational tools to design membrane proteins will transform the platform to engineer membrane proteins for therapeutic, sensor, and separation processes. While soluble protein design has advanced, membrane protein design remains challenging due to the difficulties in modeling the lipid bilayer. Electrostatics plays an intimate role in the physics of membrane protein structure and function. However, accurately capturing electrostatic energies in the low-dielectric membrane often requires expensive calculations that are not scalable. In this work, we contribute a fast-to-compute electrostatic model that considers different

the benchmark tests are available on Git Hub (https://github.com/Graylab/Implicit-Membrane-Energy-Function-Benchmark).

**Funding:** RS and JJG are supported by the National Institute of Health through grant R35GM141881-03. The funders had no role in study design, data collection and analysis, decision to publish, or preparation of the manuscript.

lipid bilayers and their features, making design calculations tractable. We demonstrate that the updated energy function improves the calculation of the tilt angle of membrane proteins, stability, and confidence in designing charged residues.

This is a *PLOS Computational Biology* Methods paper.

## Introduction

Electrostatic interactions are essential in membrane protein (MP) stability, structure, and function. Such interactions determine the way macromolecules interact with membranes: for example, adsorption and internalization of small peptides such as melittin and indolocidin are dependent on the lipid surface charge; [1] signaling and ion-transport through ion channels and active transporters are dependent on the electrostatic potential across the cellular membrane; [2] and association of many peripheral MPs involves electrostatic interactions with the membrane. [3, 4] Particularly, the lipid head groups can be charged, and the hydrophobic core of the bilayer can enhance interactions between charged particles giving rise to biophysical features that are less frequently observed in the aqueous phase. [5] Hydrophobicity scales the propensity of a given residue to be present in the hydrocarbon region. [6, 7] However, some show unusual behavior: for example, residues Arg and Lys snorkel out of the membrane interior to minimize the cost of inserting charged groups into the membrane; [8–10] aromatic residues Trp and Tyr mostly interact with the head-group region, but Phe is present in both membrane core as well as the head-group region. [11] Additionally, the non-polar bilayer increases the strength of traditional hydrogen bonds. [5, 12] Despite the importance of membrane electrostatic interactions, they are challenging to capture computationally, primarily due to the high computational cost incurred in sampling solvent and solute degrees of freedom.

One way for electrostatic calculations to overcome solute and solvent degrees of freedom is to treat the solvent as a continuous medium and ignore solvent-solvent interactions. A common approach in biological systems is solving the Poisson-Boltzmann (PB) equation with finite-difference methods representing lipids both explicitly and as continuum slabs; however, computational costs have been a bottleneck. [13–16] To overcome the cost, the generalized Born (GB) approximation represents the membrane environment through a switching function or heterogeneous dielectric slab. [17–19] The calculated GB energy includes a screening radius which in turn depends on its local environment and surrounding neighbors, however, it is still expensive and complicated. A third possibility is the heuristic Coulomb equation. In Rosetta soluble protein calculations, this model has yielded a significant speedup relative to GB calculations and complements explicit modeling of hydrogen bonds. [20, 21]

One approach to overcome the computational cost of lipid membrane complexity is to represent the membrane as a continuum. [22–26] To compare the computing time performances for an implicit and explicit membrane model, Ulmschneider and Ulmschneider [27] undertook a comprehensive analysis of sampling efficiency of WALP16 peptide folding using a GB based implicit membrane model and, an explicit DPPC lipid bilayer and an octane slab membrane mimic. A quantitative assessment of sampling efficiency, utilizing a wide range of performance metrics, revealed that the implicit membrane models were at least two orders of magnitude more efficient than the octane membrane mimic, the simplest of the two explicit

models. Lazaridis implicit membrane model (IMM1), [28, 29] is a Gaussian solvent-exclusion model that uses experimentally measured transfer energies for side-chain analogs in organic solvents to emulate the transfer energy into the cell membrane. IMM1 uses a water-to-membrane transition function-dependent dielectric constant to strengthen the electrostatic interactions in the membrane. IMM1 has been applied to various biomolecular modeling problems, including studies of peptides, [30] de-novo folding, [31] and de-novo design of transmembrane helical bundles. [32] However, organic solvent slabs differ from phospholipid bilayers because lipids are thermodynamically constrained to a bilayer configuration, resulting in a unique polarity gradient that influences side-chain preferences. [33, 34] Another alternative is to directly calculate amino-acid preferences by deriving statistical potentials from a database of known MP structures. Yet, statistical potentials do not capture varying physicochemical features of different membranes. [35–38]

The F19 energy function, which was inspired by IMM1, included the transfer energy of residues in the phospholipid bilayer based on the Moon and Fleming (MF) scale. [39, 40] However, F19 omitted the electrostatic interactions of the membrane environment, leading to errors in benchmarking tests. For example, in a test to calculate the tilt angle of adsorbed peptide cecropin A$(1 - 8)$-magainin $2(1 - 12)$ (PDB:1f0d), F19 oriented it at 56˚ with the membrane normal with its N-terminus Lys residues facing the aqueous phase. However, solution NMR found the tilt angle of the peptide at 90˚, which allowed favorable attractive electrostatic interactions between the cationic Lys residues and the anionic phosphate lipid head groups of DLPC. A second issue in F19 energy function was an underestimated water-to-bilayer transfer energy $\Delta G_{w,l}$ for Asp and Glu at neutral pH. The transfer energies for Asp and Glu derived from the MF hydrophobicity scale were measured at a pH of 3.8. [40] Since this pH is close to the nominal pKa values of Asp and Glu side chains, the experimental transfer energy was influenced by the presence of protonated Asp and Glu ($D^0$, $E^0$) residues, which are less polar than the deprotonated residues ($D^{-1}$, $E^{-1}$) at neutral pH. In this paper, our objective is to add the missing electrostatic features of the membrane to F19 energy function and test whether the new energy function, named Franklin2023 (F23), better captures features of MPs.

We explore three modifications to F19. First, we develop a fast-to-compute Coulomb-based electrostatics method that includes a low dielectric constant of the lipid bilayer. Second, we use a mean field-based calculation for the electrostatic potential due to the lipid membrane and water by solving Poisson's equation using a fast Fourier transform (FFT) based solver while avoiding any assumption or discrepancies due to ionic strength incurred in PB or its linearized version. [41, 42] Third, we used a modified value of $\Delta G_{w,l}$ for Asp and Glu residues at neutral pH.

Until very recently, the energy functions have been developed and tested for specialized tasks, questioning their generalizability. To confront the challenge of over-fitting and a specialized model, our group recently developed a 12-test benchmark suite that probes (1) protein orientation in the bilayer, (2) stability, (3) sequence, and (4) docking structures of membrane protein. [43] These tests form a platform to evaluate the strengths of an energy function and suggest areas of improvement. We evaluate the performance of F23 on predicting the (1) orientation of peptides in the membrane environment, (2) thermal stability due to point mutation, (3) transfer energy of peptides and (4) design evaluations. Our results have shown significant improvement. Relative to F19, F23 improves the calculation of the tilt angle of membrane proteins for 90% of WALP peptides of different lengths, 15% of transmembrane (TM)-peptides, and 25% of the adsorbed peptides. The performances for stability and design tests are equivalent to F19 and F23. Finally, we conclude the paper with perspectives on further improvements for implicit membrane energy functions.

## Methods

### Implicit model development

**Electrostatic energy term based on lipid type.**   We calculate the electrostatic effect of lipid bilayers based on all-atom molecular dynamics (MD) simulation of lipid molecules and solvents alone. We derive the electrostatic potential $\Psi(x, y, z)$ by solving Poisson's equation using a 3D-grid based FFT solver:

$$\nabla \cdot \nabla \varepsilon \Psi = \sum_{l=1}^{l=\text{lipid}} \rho_l + \sum_{w=1}^{w=\text{water}} \rho_w + \sum_{s=1}^{s=\text{salt}} \rho_s, \tag{1}$$

where $\rho_w$, $\rho_l$ and $\rho_s$ is the position-dependent charge densities of water, lipid, and salt averaged over MD trajectories, [44] and $\varepsilon = 8.854 \times 10^{-12}\,\text{CV}^{-1}\text{m}^{-1}$ is the dielectric constant of vacuum. To extract the depth-dependent $\Psi$, we use non-linear regression to fit the piece-wise function $\tilde{\Psi}_w$ as shown in Fig 1a. We derived the fitting parameters for all simulated lipid bilayers (Table A in S1 Text) as:

$$\tilde{\Psi}_w = \begin{cases} A_1 + \dfrac{A_1 - A_2}{\left\{ 1 + \exp\left( \dfrac{|z| - A_3}{A_4} \right) \right\}} & \text{for } |z| >= z_c \\[2em] C_1|z|^4 + C_2|z|^3 + C_3|z|^2 + C_4|z| + C_5 & \text{for } |z| < z_c \end{cases}$$

where $z_c$ is the maximum of the critical points for the function $\tilde{\Psi}_w$ for $|z| < z_c$. The fitting parameters of different lipid types are listed in Table A in S1 Text. The reference state of $\Psi = 0$ in water is set at depth $z \approx 40$ Å and the potential difference $\Delta\Psi$ is obtained (as shown by the dashed lines in Fig 1a). We compare the electrostatic potential due to salt and no salt in Fig 1b. As expected, the potential difference is found to be lower due to the presence of salt. The electrostatic energy due to the protein is then:

$$\Delta G_{\text{lipid}} = \sum_{r=1}^{N_{\text{res}}} \sum_{a=1}^{N_{\text{atom}}^r} \Delta\Psi(z) q_{r,a}, \tag{2}$$

where the sums are over all residues and all atoms in each residue, $q_{r,a}$ is the charge of a protein atom, and $\Psi(z)$ is the electric potential of the empty water-bilayer system as a function of membrane depth. $\Psi(z)$ is an average of $\Psi(x, y, z)$ in the other directions.

Following our previous work, [44] the MD trajectories used in this analysis were generated from all-atom molecular dynamics simulations of phospholipids, water, and salt. The simulations were performed using the NAMD molecular dynamics engine at a constant pressure of 1 atm and a temperature of 37°C. The CHARMM36 force field was used for lipids, and TIP3 model for water. See the methods section of Alford et al., [44] for further details.

**Bilayer depth-dependent dielectric constant.**   Due to the hydrophobic environment within the membrane, the dielectric constant is lower than that in the isotropic soluble region. To account for the dielectric variation, in IMM1 a water-to-membrane transition-function-dependent dielectric constant is substituted in EEF. [28]

Inspired by IMM1 in F23, for structure prediction and design, we use a membrane hydration $f_{\text{hyd}}$ dependent dielectric constant. Membrane hydration ($f_{\text{hyd}}$) is a proxy of the membrane depth and is capable of capturing water cavity effects. [44] In the center of the membrane, when an atom is exposed to lipids, $f_{\text{hyd}} = 0$; whereas when an atom is fully exposed to water, $f_{\text{hyd}} = 1.0$. When an atom is in the membrane yet faces a water cavity, $0.0 < f_{\text{hyd}} < 1.0$. For an

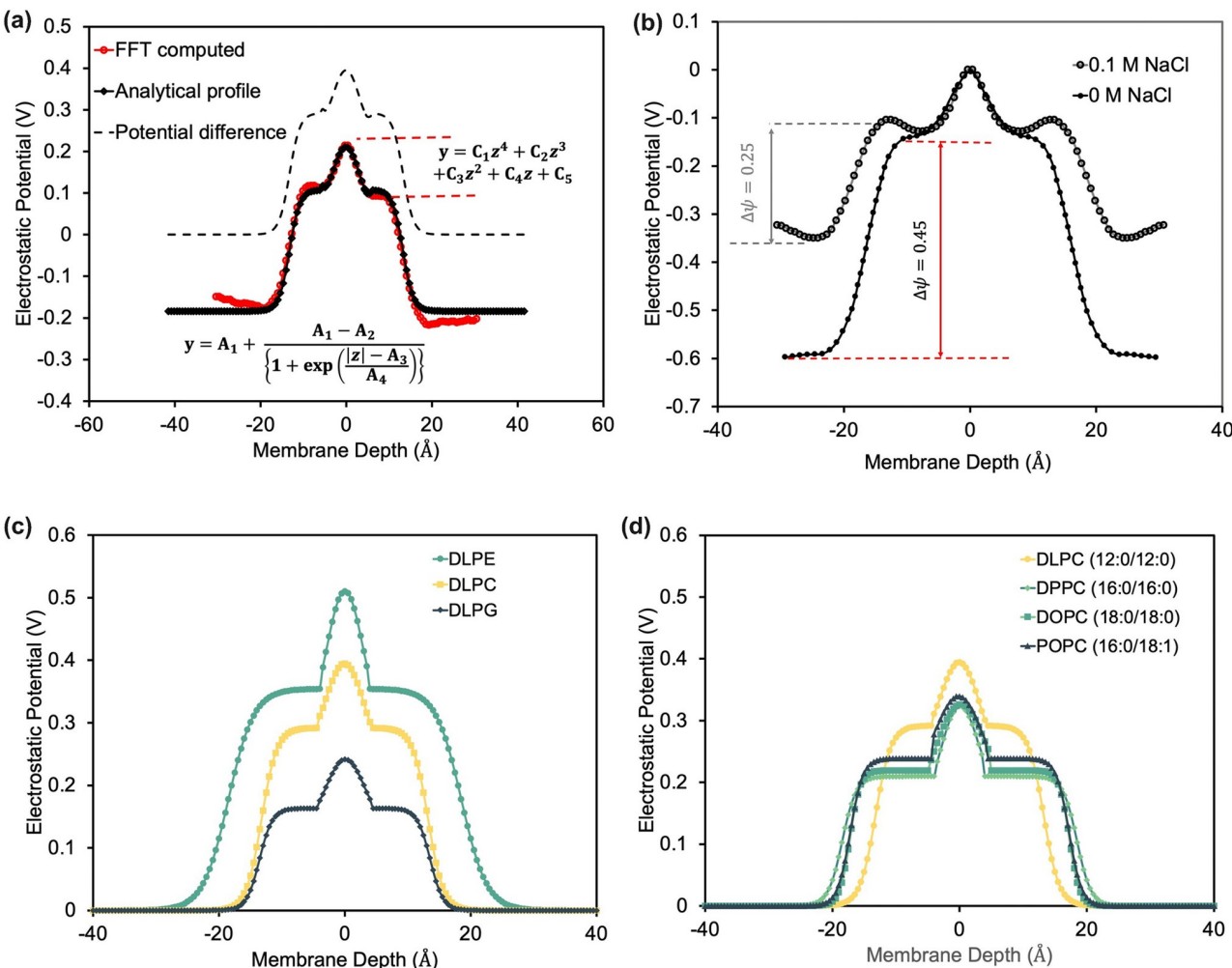

**Fig 1. Features of electrostatic potential due to the lipid bilayers in the membrane.** (a) Electrostatic potential as a function of depth and analytical fit. (b) Comparison of electrostatic potential in a 0.1 M salt solution and that calculated by Yu *et al.* in pure water. [45] Comparison of electrostatic potential as a function of membrane depth for (c) different lipid head groups and (d) lipid tail lengths.

atom pair, we define the hydration fraction $f_{\text{hyd},ij}$ as the geometric mean of $f_{\text{hyd},i}$ and $f_{\text{hyd},j}$. Rosetta uses a distance-dependent dielectric constant for soluble proteins with sigmoidal function $\varepsilon_{\text{sol}}(r_{ij})$ to transition between the protein core ($\varepsilon_{\text{core}} = 6$) and surface ($\varepsilon_{\text{surface}} = 80$). [46] To define the distance-dependent dielectric constant within the membrane, we use the same sigmoidal function $\varepsilon_{\text{memb}}(r_{ij})$ to transition between protein core ($\varepsilon_{\text{core}} = 3$) and surface ($\varepsilon_{\text{surface}} = 10$) as summarized in Fig 2a. [47, 48] To combine the effect of distance dependence and membrane hydration, we use a linear mixture equation:

$$\varepsilon(r_{ij}, f_{\text{hyd},ij}) = f_{\text{hyd},ij}\varepsilon_{\text{sol}}(r_{ij}) + (1 - f_{\text{hyd},ij})\varepsilon_{\text{memb}}(r_{ij}) \tag{3}$$

Fig 2b shows $\varepsilon(r_{ij}, f_{\text{hyd},ij})$ as a function of $f_{\text{hyd},ij}$ and atom pair distance $r_{ij}$. Based on the modified dielectric constant, the Coulomb energy is:

$$E_{\text{elec},ij}^{\text{dielectric}} = \frac{C_0 q_i q_j}{\varepsilon(r_{ij}, f_{\text{hyd},ij})} \left[ \frac{1}{r_{ij}} - \frac{1}{r_{\text{max}}} \right], \tag{4}$$

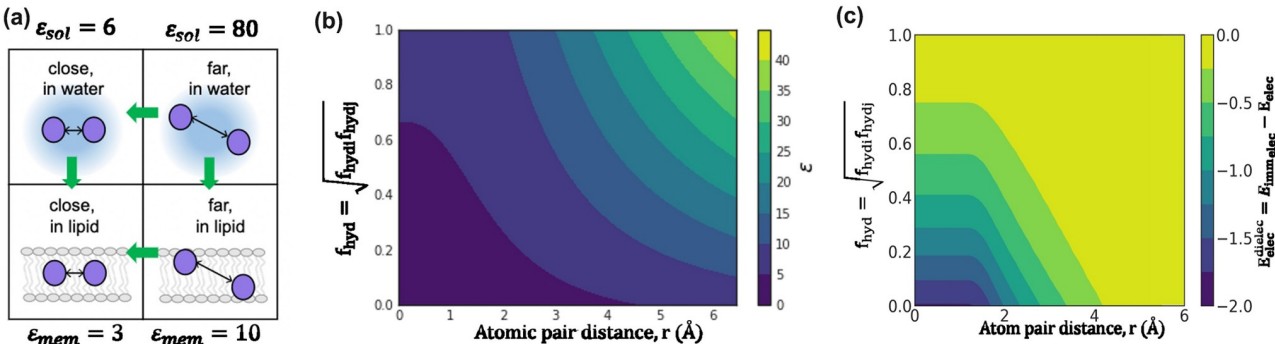

**Fig 2. Coulomb electrostatic energy due to the lipid bilayers in the membrane.** (a) Membrane-dependent electrostatics potential: The energy increases when entering the bilayer and at close atom pair distances. Green arrows indicate the direction of increasing energy. (b) Dependence of the dielectric constant $\varepsilon$ on fraction of hydration $f_{\mathrm{hyd}}$ and atom-pair distance $r_{i,j}$. The different dielectric constants are colored from low (dark blue) to high (yellow). (c) Variation of the membrane dielectric-dependent electrostatics energy as a function of fractional hydration $f_{\mathrm{hyd}}$ and atom-pair distance $r_{i,j}$. Each grid point corresponds to an energy calculated for point charges with opposing signs, and the energy varies from strong (dark blue) to weak (yellow).

where $q_i$, $q_j$ are partial charges of protein atoms, $C_0$ is Coulomb's constant (322 Å kcal/mol $e^{-2}$), where $e$ is the elementary charge and the electrostatic energy is truncated at $r_{\mathrm{max}} = 5.5$ Å. [39] To isolate the excess electrostatic energy due to the membrane, in this energy term, we subtract the electrostatic energy from the solution. In addition, nullifying the electrostatic energy ensures the reference energy of the unfolded protein in the soluble region remains unaffected. [49] The modified energy is given as:

$$\Delta E_{\mathrm{elec},ij}^{\mathrm{dielectric}} = C_0 q_i q_j \left[ \frac{1}{r_{ij}} - \frac{1}{r_{\mathrm{max}}} \right] \left[ \frac{1}{\varepsilon(r_{ij}, f_{\mathrm{hyd},ij})} - \frac{1}{\varepsilon(r_{ij})} \right] \tag{5}$$

Fig 2c presents the electrostatic energy due to the membrane as a function of pair fraction of hydration $f_{\mathrm{hyd},ij}$ and atom pair distance.

**Modifying $\Delta G_{w,l}$ for Asp and Glu at neutral pH.** F19 included the hydrophobicity of residues as per the Moon and Fleming (MF) hydrophobicity scale, [40] which reflects the water-to-bilayer partitioning of side chains in the context of a native transmembrane protein spanning a phospholipid bilayer. The MF hydrophobicity scale is linearly correlated to Wimley White (WW) scale with a higher slope, which is based on side-chain water-to-octanol transfer energies (as seen in Fig 3a). [40, 50] The two exceptions are Asp and Glu since the MF partition energy is measured at a pH close to their $\mathrm{pK_a}$. The question arises, how do we calculate the values of Asp and Glu at neutral pH ($D^{-1}$, $E^{-1}$)? To determine the MF-scale transfer energy of $D^{-1}$, $E^{-1}$, we use the linear best fit of MF and WW hydrophobicity scales and extrapolate the WW transfer energy of $D^{-1}$ and $E^{-1}$.

Based on the MF scale transfer energy of $D^{-1}$, $E^{-1}$, we used linear regression to calculate the transfer energy of atoms $\Delta G_{w,l}^{\mathrm{atom}}$ by solving $[\mathbf{A}_{\mathrm{aa,atom}}] \cdot \Delta \mathbf{G}_{w,l}^{\mathrm{atom}} = \Delta \mathbf{G}_{w,l}^{\mathrm{aa}}$, where $[\mathbf{A}_{\mathrm{aa,atom}}]$ is a matrix of atom type stoichiometry coefficients and the $\Delta \mathbf{G}_{w,l}^{\mathrm{atom}}$ and $\Delta \mathbf{G}_{w,l}^{\mathrm{aa}}$ represent the vectors of $\Delta \mathbf{G}_{w,l}$ for different atom and amino acid types. We append $[\mathbf{A}_{\mathrm{aa,atom}}]$ and $\Delta \mathbf{G}_{w,l}^{\mathrm{aa}}$ to include the stoichiometry of $D^{-1}$ and $E^{-1}$ to recalculate $\Delta \mathbf{G}_{w,l}^{\mathrm{atom}}$. Table B in S1 Text shows all the equations and values of $\Delta \mathbf{G}_{w,l}^{\mathrm{aa}}$ we have used. Fig 3b compares $\Delta \mathbf{G}_{w,l}^{\mathrm{atom}}$ calculated with and without $\Delta \mathbf{G}_{w,l}^{\mathrm{aa}}$ for $D^{-1}$ and $E^{-1}$ used in F23 and F19 respectively. The additional rows for $D^{-1}$ and $E^{-1}$ led to renewed values of $\Delta G_{w,l}^{\mathrm{COO}}$ and $\Delta G_{w,l}^{\mathrm{OOC}}$ as seen in Fig 3a and 3b.

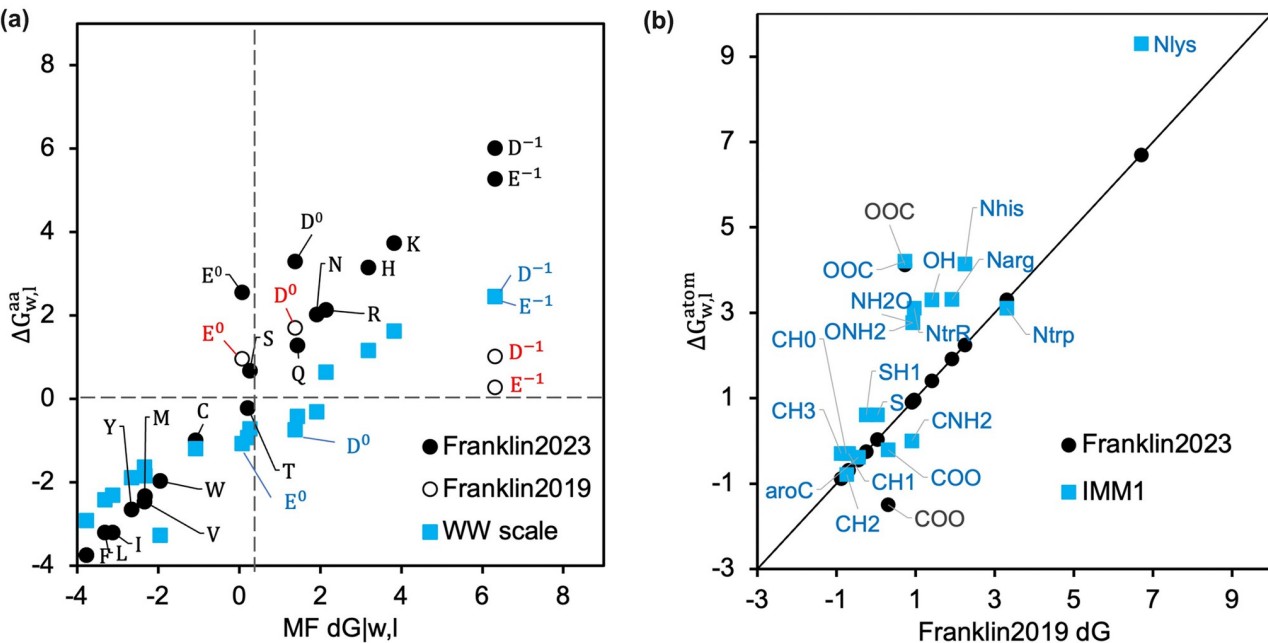

**Fig 3. Values of $\Delta G_{w,l}^{\text{atom}}$ and $\Delta G_{w,l}^{\text{aa}}$ with different scales.** (a) Comparison of $\Delta G_{w,l}^{\text{aa}}$ calculated using F19 and F23 relative to the experimental values from MF and WW scales. The MF $\Delta G_{w,l}^{\text{aa}}$ for $D^{-1}$ and $E^{-1}$ are estimated by extrapolation of a linear fit of the other WW data points. (b) Comparison of $\Delta G_{w,l}^{\text{atom}}$ calculated using F23 and IMM1 relative to F19.

## Integration of new terms into F23

We integrate two new energy terms ($\Delta G_{\text{lipid}}$ and $\Delta E_{\text{elec},ij}^{\text{dielectric}}$) to F19 through linear combination of terms.

$$\text{F23} = \text{Ref2015} + w_{w,l}\Delta G_{w,l} + w_{\text{lipid}}\Delta G_{\text{lipid}} + w_{\text{dielec}}\Delta E_{\text{elec},ij}^{\text{dielectric}}, \tag{6}$$

where Ref2015 is Rosetta's standard energy function for soluble proteins, [46]$\Delta G_{w,l}$ add the transfer energy from water to bilayer, [44] and $\Delta G_{\text{lipid}}$ and $\Delta E_{\text{elec},ij}^{\text{dielectric}}$ are the energies due to the lipid head group and the membrane dielectric constant. We can recover the energy function F19 by setting $w_{w,l} = 0.5$, $w_{\text{lipid}} = 0$ and $w_{\text{dielec}} = 0$. We solved the simultaneous equations for the three unknown positive weights to maximize the correlation coefficient for the experimentally measured $\Delta\Delta G_{\text{mut}}$ for amino acids at the center of OmpLA protein [40] and mutations to aromatic side chains (Phe, Trp, Tyr) [51]. The resulting weights were $w_{w,l} = 1$, $w_{\text{lipid}} = 0.128$ and $w_{\text{dielec}} = 0.01$, achieved using L-BFGS-B algorithm within bounds of [0, 1]. Our choice of these bounds was to avoid negative weights and extremely large positive weights while recognizing potential biases due to the bounds and more data points at the center of the membrane. We omitted the matrix rows for Ala due to its role as a host residue for all experimental mutations, Pro to avoid steric clashes resulting in large values, Asp and Glu residues because the experimental values were measured at low pH.

## Results and discussion

### Biologically realistic membrane features

**F23 captures the effects of the diverse lipid bilayer.** We first investigate the effect of the lipid head and tail group (Fig 1c and 1d) by examining the electrostatic potential of the empty

lipid layer in the presence of counter-ions and salt solution. While this implicit method cannot capture the interface deformation or interaction of lipids with the residues, this electrostatic potential will be sensitive to different lipid types. Fig 1c shows the $\Delta\Psi$ for three different lipid head groups (DLPE, DLPC, and DLPG) with the same tail group, *i.e.* Lauric acid(12: 0) acyl chains (DL). Phosphatidylethanolamine (PE) and phosphatidylcholine (PC) have neutral head groups with the same overall composition, except that PE has a $N^+H_3$ and PC has a $N^+(CH_3)_3$ atom. Relative to PC, the smaller head group of PE increases the local charge density leading to higher electrostatic potential. [52, 53] Phosphatidylglycerol (PG), on the other hand, has anionic lipid head groups known to adsorb counterions to the bilayer interface strongly, screening the net electrostatic potential relative to PC. [54, 55] Fig 1d shows the $\Delta\Psi$ for four different lipid tail groups (DLPC, DPPC, DOPC, and POPC) with the constant head group PC and different length and degree of unsaturation. These calculated results match chain length trends observed in experimentally measured surface potential. [56] Using these different profiles, F23 can approximate different homogeneous lipid environments through the potential $\Delta\Psi$, and parameters for other mixed lipid compositions can be added with a short MD run.

## Benchmark performance of F23

We evaluate the performance of F23 using five benchmark tests. The tests are designed to evaluate an energy function's ability to replicate measured membrane protein stability and perform structure prediction and design. We compare the performance of F23 to two other existing implicit models: (1) F19 and (2) M12, which is based on IMM1. [39, 57] We chose these models for their low computational cost which allows evaluation with structure prediction and design tests.

**Test 1: Orientation of polyalanine and WALP peptides.** MPs are thermodynamically stable in the bilayer because of favorable orientation and insertion energy. Thus, a key challenge for energy function and the focus for our first and second tests is to recapitulate this lowest energy orientation of membrane peptides. Following our prior work [43, 44, 58], we used a protocol that samples all possible orientations of the peptide relative to the implicit membrane within $\pm 60$Å of the bilayer center (d), tilt angles relative to the membrane normal ($\theta$) between $\pm 180°$, rotation angles relative to the principal helical axis ($\phi$) between $0 - 360°$. The global energy minimum of all sampled positions is defined as the most stable predicted orientation. In this first test, we test the effect of the length of repeating helical peptides on the tilt angle.

**Polyalanine**. To test the effect of peptide length, we predict the tilt angle of polyalanine peptides with a varying number of alanine residues (ranging from 20–40). Polyalanine peptides have been shown to adopt predominantly helical conformations in lipid bilayers. We model the residues as $\alpha$–helices with $\psi = -47°$ and $\phi = -57°$ with charged ends. We compare our results with orientations calculated using Sengupta *et al.*'s 5-slab continuum dielectric model with CHARMM36 force field [59] as shown in Fig 4a. Due to the lack of any lipid type mentioned in the 5-slab model, in this test, we calculated the tilt angles using DLPC for both F19 and F23. Similar to that in Sengupta *et al.*, [59] the tilt angles calculated by F23 and F19 are proportional to the length of the peptide. However, relative to the 5-layer slab model, F19 and F23 over-predict the tilt angle by about 10–15°. Irrespective of the size of the polyalanine, M12 places the peptides outside the membrane at a tilt near 90°.

**WALP peptides**. WALP peptides typically comprise two Trp residues at each end with a helical core composed of alternating Leu and Ala residues: GWW(LA)$_n$LWWA. They have proven to be opportune models for experimentally investigating lipid and peptide interactions. [60] The WALP peptides are modeled as ideal $\alpha$-helices with $\psi = -47°$ and $\phi = -57°$. In this test, we calculate the tilt angle of WALP peptides of different lengths in DMPC lipid layers

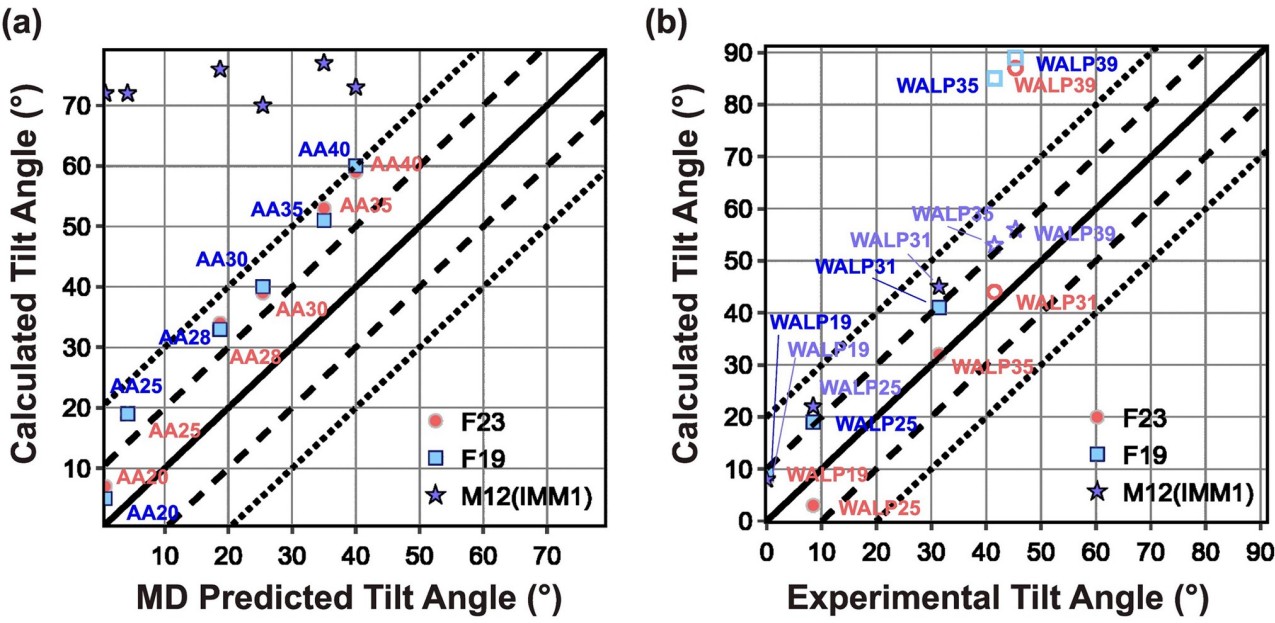

**Fig 4. Test1: Orientation of polyalanine and WALP peptides.** (a) Predicted tilt angle of polyalanine as a function of the number of residues. The polyalanine is named AAx, where x represents the number of residues. (b) Predicted tilt angle of WALP peptides as a function of the number of residues. The WALP peptides are presented as WALPx, where x represents the number of residues. Peptides for which experimental results are available are presented by filled markers, and those for which simulated data is available are shown by unfilled markers.

(using M12, F19, and F23). We compare the results with experimental solid-state NMR (ssNMR) and spectroscopic measurements available for WALP19–31 (shown in Fig 4b by filled markers) and the rest of the results predicted by Sengupta *et al.*'s 5-slab model (shown in Fig 4b by unfilled markers). [59, 61, 62] Sengupta *et al.* calculated the tilt angle of fixed backbone peptide by determining the energy landscape as a function of depth and tilt angle that included the peptide energy using CHARMM potential and membrane solvation energy using electrostatics and non-polar terms.

Similar to ssNMR, F19 calculated the tilt angle of WALP to be directly proportional to the number of residues (Fig 4b). For shorter peptides, the tilt angle is in good agreement with the experiment; however, for the peptides longer than WALP31, F19 buries the peptides at the center of the membrane (tilt angle 90˚). For longer peptides, F19 is unable to compensate the net hydrophobicity due to the non-polar residues by the energy due to polar residues, resulting in burying the peptide at the center of the membrane (see examples of WALP35 and WALP39 in Fig A(a-b) and A(d-e) in S1 Text). To demonstrate the impact of lipid composition, Fig E in S1 Text displays calculated WALP tilt angles using F23 in three different lipid compositions.

Relative to F19, F23 computed the tilt angle for WALP peptides closer to the experimental measures and those predicted by the 5-slab model. Particularly, F23 increased the accuracy of tilt angle prediction due to two factors: (a) lower hydrophobicity of the polar residues at the terminal (Fig A(c) in S1 Text), and (b) the additional membrane electrostatic energy. However, F23 still buried WALP39 with a tilt angle much higher than the angle predicted by the 5–slab model (Fig A(d) in S1 Text). Contrary to F23 and F19, M12 calculated the tilt angle of all peptides close to those by the 5–slab model (Fig 4b) because both the models use the WW hydrophobicity scale, which has a lower slope than the MF hydrophobicity scale used by F19 and F23 (Fig 3a). [40, 43] However, hydrophobicity alone cannot be the reason, as otherwise, the tilt angles for polyalanines would have been similar. M12 uses a knowledge term based on the

probability of finding a residue at a given depth of the membrane, which penalizes the Trp residues in the membrane but not the Ala residues. This heuristic term correlates well with the location of energy minima of the total energy Fig B(a-b) in S1 Text.

In summary, with the combination of hydrophobic transfer energy and electrostatic interaction energy with the membrane relative to F19, F23 improves the accuracy of tilt angle calculation. However, F23 buries long WALP peptides due to higher hydrophobic energy relative to M12 (IMM1).

**Test 2: Orientation of TM- and adsorbed peptides.** Our second test also concerns the tilt angle for peptides. However, unlike Test 1, which explored related families in a membrane, these peptides are of diverse sequences and in different lipid environments, with some that adsorb on the surface of the membrane and some that pass through it. The different lipid types used for experiments and the parameters used for this test are listed in Table C in S1 Text. The objective here is to see if the lowest energy orientation of the membrane peptides identified by the energy function corresponds to the native orientation. This test is the cornerstone of our benchmark because it was used for the validation of early implicit membrane models. [43, 63] The tilt angles of TM-peptides were measured previously using solid-state NMR experiments using different lipid compositions, including DPC micelles, DMPC vesicles, mixed DOPC: DOPG bilayers, and pure DOPC bilayers. [43, 63] While the experimental uncertainty is not available for all measurements, tilt angle measurements have a typical error range of ±3–5˚. The tilt angle for absorbed peptides is measured by solution NMR in dodecyl phosphocholine micelles or trifluoroethanol with an uncertainty of ±6–12˚. [64] Since our previous publications have already reported a comparison of F19 with M12 and other energy functions, we focus here on the comparison of F23 with F19 (Fig 5a and 5b). [43]

F19 calculated the tilt angle within ±20˚ of the experimentally measured values for three out of four peptides. F23 improved the tilt angle calculation for WALP23 by 20˚ and NR1 subunit of the NMDA receptor (2nr1) peptides by 5˚ to result in tilt angles within ±20˚ of the

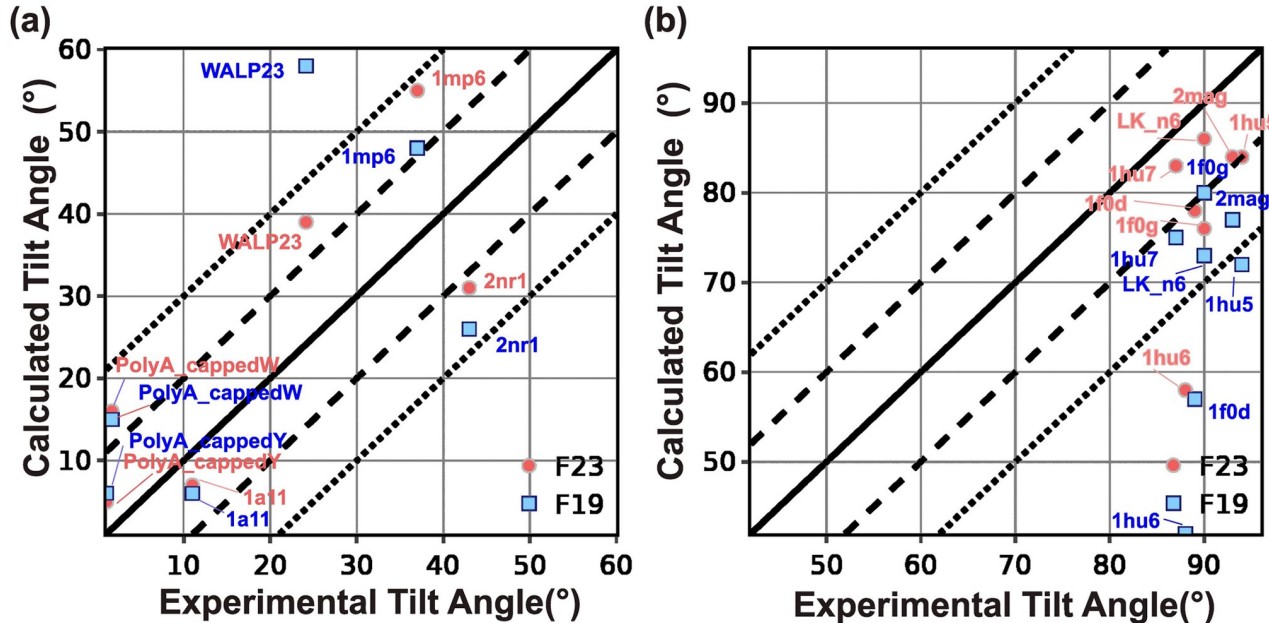

**Fig 5. Test2: Orientation of TM and adsorbed peptides.** (a) Predicted tilt angle of TM peptides compared with experimental measurements. (b) Predicted tilt angle of adsorbed membrane peptides and compared with experimental measurements.

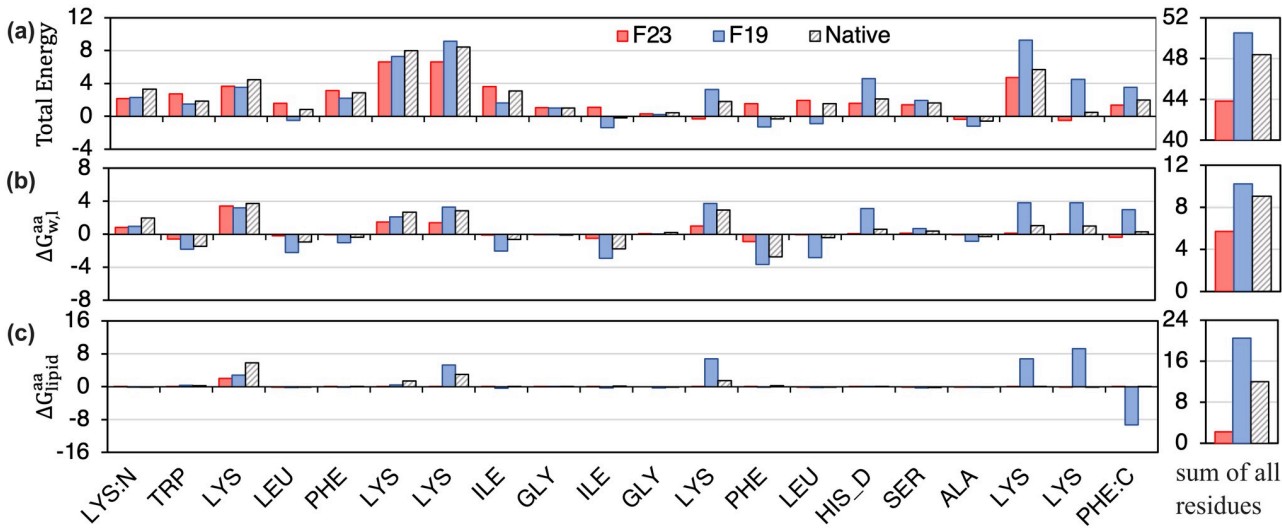

**Fig 6. Per-residue F23 energy calculated at native, F23 and F19 calculated tilt angles.** (a) F23 per-residue total energy (REU), (b) per-residue transfer energy to the bilayer, and (c) per-residue electrostatic energy due to the lipid layer at native (black hashed), F23 (red) and F19 (blue) calculated tilt angles. **Right**: Sum of all residues.

experimentally measured values for all the four peptides (as seen in Fig 5a. F23 worsened the tilt angle for influenza A M2 (1mp6) peptides.

To understand the deviation between the experiment and calculated tilt angles of WALP23 and 1mp6, we examine the energy landscape under F23 and F19. F23 improved the tilt angle of WALP23 due to the lower hydrophobicity of the C-terminus. To illustrate the reasons for the difference in tilt angle for 1mp6, we present the energy landscape under F23 and F19 in Fig C(a-g) in S1 Text. F19 calculated the tilt angle at $z = -5$ Å as 48°. Due to the lower hydrophobicity of C-terminus Asp and Leu residues, F23 further improved the tilt angle of 1mp6 at $z = -5$ Å(Fig C(f) in S1 Text). However, electrostatic interactions between arginine and the lipid layer reduced the energy of F23 further by 0.1 REU and led to a worsening of the calculated tilt angle (Fig C(f-g) in S1 Text).

For four out of seven adsorbed membrane proteins, F19 calculated the tilt angle within ±20° of the experimentally measured values. F23 significantly improved calculation resulting in tilt angles within ±20° of the experimentally measured values for six out of seven peptides (Fig 5b). F23 reduced the error in tilt angle of magainin-cecropin hybrid (pdb:1f0d) and novis-pirin G10 (pdb:1hu6) by 21° and 16° respectively.

We compare the F23 per-residue energy of the magainin-cecropin hybrid (pdb:1f0d) at configurations predicted by F19, F23, and the native state (Fig 6b). F23 penalizes Lys residues at the tilt angle calculated by F19 and allows favorable attractive electrostatic interactions between the cationic side chains and the anionic phosphate lipid headgroups of DLPC. In addition, F23 assigned higher energy to the polar Lys residues present near the center of the membrane in the F19 configuration relative to the surface absorbed native state. To further establish the importance of the electrostatic interaction, we recalculate the residue with a special version of F23 (shown in Fig D(d-f) in S1 Text as dashed lines) in which the $\Delta G_{w,l}^{atom}$ parameters are reverted back to those of F19. Indeed, higher electrostatic repulsion at the center results in the magainin-cecropin hybrid peptide (pdb:1f0d) being placed near the surface of the membrane and obtaining a near-native tilt angle, confirming that the $\Delta G_{lipid}$ and $\Delta E_{elec,ij}^{dielectric}$ terms are responsible for the improvement.

To summarize, the tilt angle calculation by F23 for several TM-peptides and adsorbed peptides has improved due to the addition of the new electrostatic terms and the renewed values of $\Delta G_{w,l}^{\mathrm{COO}}$ and $\Delta G_{w,l}^{\mathrm{OOC}}$.

**Test 3: $\Delta\Delta G_{\mathrm{mutation}}$.**    Our third test evaluates how a change in the sequence of a membrane protein affects its overall thermostability. $\Delta\Delta G_{\mathrm{mutation}}$ is the change in Gibbs free energy of folding from water to the lipid bilayer of each mutant relative to the native sequence.

$$\Delta\Delta G_{\mathrm{mutation}} = [G_{\mathrm{bilayer}}^{\mathrm{mutant}} - G_{\mathrm{water}}^{\mathrm{mutant}}] - [G_{\mathrm{bilayer}}^{\mathrm{native}} - G_{\mathrm{water}}^{\mathrm{native}}], \tag{7}$$

To design MPs, $\Delta\Delta G_{\mathrm{mutation}}$ is crucial in evaluating the effect of genetic mutations on protein functions. To evaluate $\Delta\Delta G_{\mathrm{mutation}}$, we use a Rosetta fixed-backbone and fixed-orientation protocol. [44] We use two sets of $\Delta\Delta G_{\mathrm{mutation}}$ measurements from the Fleming lab taken at equilibrium in DLPC vesicles and in the context of a $\beta$-barrel protein scaffold. The two datasets are mutations from Ala to all 19 remaining canonical amino acids at position 210 on outer membrane phospholipase A (OmpLA), [40] and at position 111 on an outer membrane palmitoyl transferase (PagP). [51] The experimental uncertainty of the measured $\Delta\Delta G_{\mathrm{mutation}}$ values is $\pm 0.6$ kcal mol$^{-1}$.

Fig 7a and 7b compares the experimental and calculated $\Delta\Delta G_{\mathrm{mutation}}$ from Ala to all other residues for OmpLA and PagP proteins at a fixed height in the membrane. We also included $D^{-1}$ and $E^{-1}$ to emulate these residues at neutral pH. While F19 linearly correlates well for both OmpLA and PagP, F23 predicts the $\Delta\Delta G_{\mathrm{mutation}}$ closer to the experimental values (*e.g.* in OmpLA H improves from 7.2 to 5.2 REU, K from 6.4 to 5.2 REU, N from 6.2 to 4.7 REU; in PagP R improves from 5.9 to 3.8 REU and K from 8.5 to 7 REU), evident from the slope of the

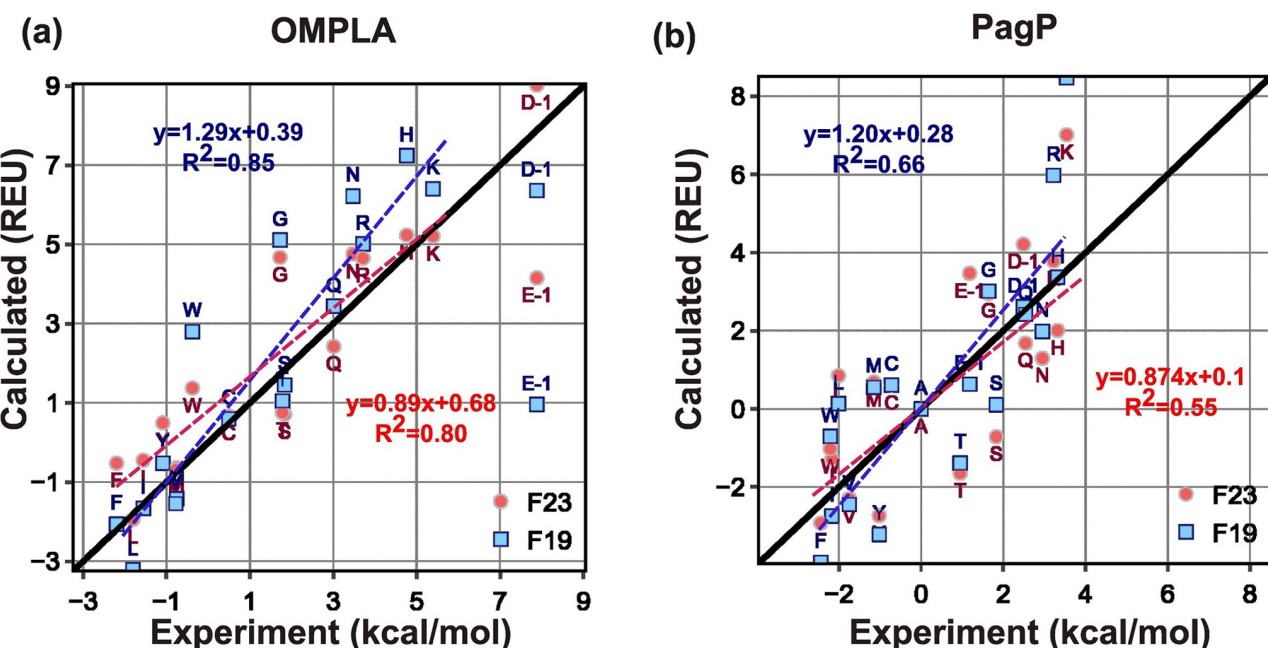

**Fig 7. Test3: $\Delta\Delta G_{\mathrm{mutation}}$ of alanine to all other residues.** Predicted $\Delta\Delta G_{\mathrm{mutation}}$ and experimental measurements for (a) OmpLA and (b) Pagp protein. For the experimental values of $\Delta\Delta G_{\mathrm{mutation}}$ for $D^{-1}$ and $E^{-1}$ for OmpLA, we have used their extrapolated MF scale values as described in Methods section. Since the experimental $\Delta G_{w,l}^{\mathrm{aa}}$ based on the MF hydrophobicity, was measured at a pH of 3.8, where some of the Asp and Glu are in their protonated states, we removed them while calculating the slope and correlation coefficient of the best-fit lines (dashed lines) for the predictions relative to experimental values. The black line is the y = x line. The red and blue dashed lines are linear best-fit lines for values predicted by F23 and F19, respectively. The equations for the best-fit lines for F23 and F19 are shown in red and blue, respectively.

  

best-fit line (the fit excludes Asp and Glu, see the caption for further details). To assess the uncertainty of the predicted $\Delta\Delta G_{\text{mutation}}$, we conducted 10 repeated calculations, resulting in standard deviations of 0.008 (F23, OmpLA), 0.016 (F23, PagP), 0.086 (F19, OmpLA) and 0.009 (F19,PagP).

**Test 4: $\Delta\Delta G_{\text{ins}}$.**   Next, we asked whether the energy function can recapitulate the thermodynamics of protein insertion into the membrane. This test set includes five poly-Leu peptides designed by Ulmschneider *et al.*, [65] four of which follow a $GL_XRL_XG$ motif, where $X = 5, 6, 7, 8$. The fifth follows a different pattern by adding a flanking Trp: $GWL_6RL_7G$. We compare our results against transfer energies from MD simulations in POPC bilayers, which were validated against intrinsic fluorescence measurements. The experimental uncertainty of the measured $\Delta\Delta G_{\text{ins}}$ values was $\pm 1.4$ kcal mol$^{-1}$. [65]

To evaluate the performance of energy functions, we compute the $\Delta\Delta G_{\text{ins}}$ (Eq 8) as the energy difference between the lowest energy orientation of the peptide in the lipid bilayer phase and in the aqueous phase:

$$\Delta\Delta G_{\text{ins}} = [G_{\text{lipid}} - G_{\text{ref}}] - [G_{\text{water}} - G_{\text{ref}}], \tag{8}$$

where $G_{\text{ref}}$ is the Gibbs free energy of an unfolded protein in solution.

Fig 8 compares $\Delta\Delta G_{\text{ins}}$ calculated by F19 and F23 to that by MD simulation. The correlation coefficient for F23 is 0.01 higher than F19, which is insignificant. However, the slopes for the best-fit lines are about double for F23 relative to F19. Relative to F19, the additional electrostatic terms in F23 increased favorable energy due to insertion. F19 and F23 recognize the relative $\Delta\Delta G_{\text{ins}}$ among the design peptides; however, the energy of insertion is overestimated

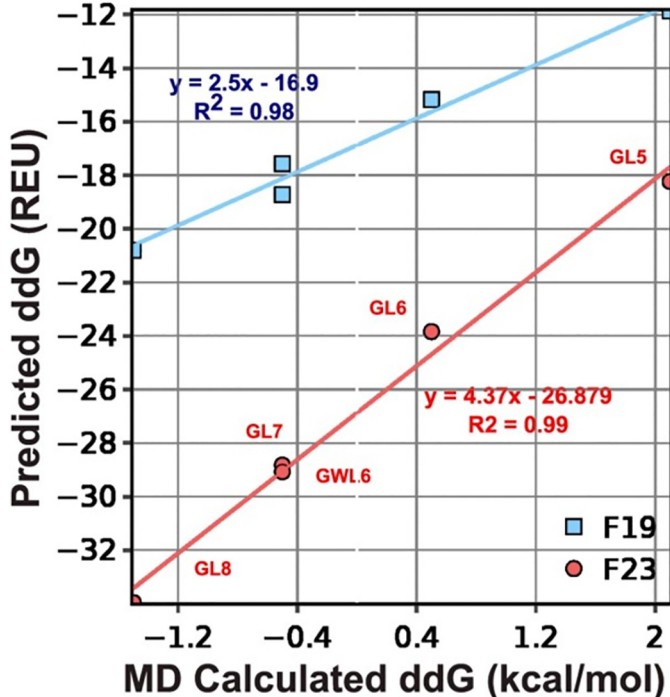

**Fig 8. Test4: $\Delta\Delta G_{\text{ins}}$ of polyleucines.** $\Delta\Delta G_{\text{ins}}$ is the energy difference between the lowest energy orientation of the peptide in the lipid bilayer phase and that in the aqueous phase. $\Delta\Delta G_{\text{ins}}$ is a measure of the affinity of a folded protein to be in the membrane phase relative to the aqueous phase. Predicted $\Delta\Delta G_{\text{ins}}$ are compared with experimental and values calculated by MD simulations.

  

compared to the calculations from MD. As in many other Rosetta-based calculations, these energy functions reproduce trends but not the exact values.

**Test 5: Sequence recovery.**   Finally, for the design applications, we evaluate the ability of an energy function to recover native MP sequences. We perform redesign using Rosetta's Monte Carlo fixed-backbone protocol that samples possible sequences using a full protein rotamer-and-sequence optimization and a multicool-simulated annealing. [66] Each protein is initialized and fixed in the orientation from the OPM database. [67] For each backbone conformation, we generate 50 designed sequences and chose the lowest scoring sequence for further analysis. We compute three metrics: (1) sequence recovery, which measures the fraction of native amino acids recovered in the redesigned sequence, (2) Kullback–Leibler (KL) divergence ($D_{KL}$), which measures the difference in the probability distribution of design and native residues, and (3) perplexity, which measures the confidence of a model in the designed sequences. To extract features relevant to the membrane, we compute these metrics for subsets of amino acids exposed to the aqueous phase (outside of membrane or pore facing), lipid phase, and interface region. The dataset for the test includes 204 MPs, including both $\alpha$-helical bundles and $\beta$-barrels, with all entries having a resolution of 3.0 Å or better and less than 25% sequence identity. The native and host lipid compositions for these proteins vary widely and include compositions not yet covered by the F23 lipid parameters; thus, for simplicity, we perform all design calculations in the DLPC bilayer.

High sequence recovery has long been correlated with strong energy function performance for soluble proteins. [49] F19 designed sequences are 31.8% identical to their native sequences (Fig 9a). With the new electrostatics terms, F23 recovered 30.8% of native residues. To estimate the variation of designed sequences for a given backbone, Fig F(a-b) in S1 Text shows the mean and standard deviation of sequence recovery and total score of all 50 designs for top 20 performing PDB backbones. The soluble protein energy function (R15 [46]) and other previous implicit membrane energy function, which was parameterized from the behavior of side-chain analogs in organic solvents (M07 [31]), lagged in sequence recovery. To compare the influence of different solvent environments, we calculated sequence recovery over subsets of residues. First, we compare buried vs. solvent-exposed side chains (Fig 9b). Among buried side chains, F19 recovered 3.4% higher native residues than F23. On the surface, sequence recovery for M12 and F23 was 3% higher than F19. This improvement in the sequence recovery of surface-exposed residues in contrast to those buried by F23 may indicate a potential path for improvement that might include higher weights of the dielectric constant term. F23 recovered a higher fraction and more variety of native residues (above random selection) relative to F19 for both water-facing and lipid-facing residues (Fig 9c and 9d).

**Amino acid distribution in design proteins is similar to that in native proteins**. To evaluate the distribution of amino acid types in the designed proteins relative to that in native proteins, we calculated $D_{KL}$ as:

$$D_{KL} = \sum_{s \in AA_{20}} p_s^{des} \log\left(\frac{p_s^{des}}{q_s^{nat}}\right),$$  (9)

where $p_s^{des}$, $q_s^{nat}$ are the probabilities of amino acid type $s$ in the design and native sequences respectively and $AA_{20}$ is the set of all 20 canonical amino acids. $p_s$ is calculated as:

$$p_s = \frac{\sum_{i=1}^{N_{bb}} \sum_{j=1}^{N_i^{res}} \mathbb{1}_{x_{i,j}}(s)}{\sum_{i=1}^{N_{bb}} N_i^{res}}$$  (10)

where $N_{bb}$ is the total number of protein backbones in our design set, $N_i^{res}$ is the length of the

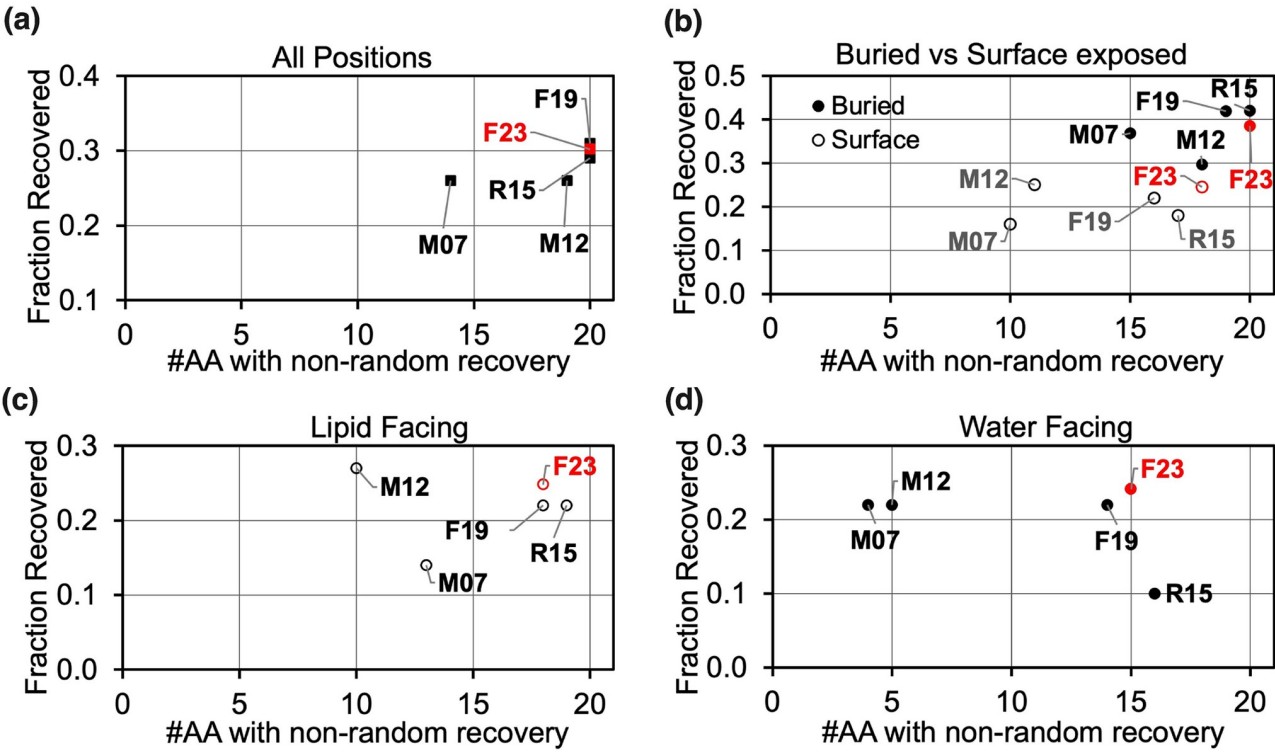

**Fig 9. Test 5: Sequence recovery of membrane proteins.** Plots show the fraction of native amino acids recovered on the y-axis and the fraction of amino acid types with individual recovery rates greater than 0.05 on the x-axis. An accurate energy function would have a high sequence recovery rate both overall and for the individual amino acid types. The results are shown for all positions in panel (a), buried (in filled circles) vs. surface-exposed(in open circles) positions in (b), lipid-exposed positions in (c), and water exposed in (d).

$i^{\text{th}}$ protein, $x_{i,j}$ is the amino acid type of the $j^{\text{th}}$ residue in the $i^{\text{th}}$ protein and, $\mathbb{1}_{x_{i,j}}(s):=1$ if $x_{i,j} = s$ or 0 otherwise.

For residue level understanding, we calculate $D_s = \log\left(\frac{p_s^{\text{des}}}{q_s^{\text{nat}}}\right)$, which is the difference in log-likelihood of amino acid type $s$ in designed and native sequences, and their sum $D = \sum_{s \in \text{AA}_{20}} D_s$. A negative $D_s$ indicates that the amino acid $s$ is under-enriched in designed sequences, and a positive $D_s$ indicates that $s$ is over-enriched. An ideal energy function will have $D_{\text{KL}} = D_s = D = 0$.

F19 outperforms Ref15 and other implicit energy functions M12 and M07 with a low and negative divergence $D = -2.7$ and $D_{\text{KL}} = 0.06$. F23, on the other hand, under-performed F19 with a $D = -4.8$ and $D_{\text{KL}} = 0.27$, which is still lower than previous energy functions (Fig 10a and 10b). Both F19 and F23 can design proteins with amino acid distribution comparable to native MPs. However, with the additional electrostatic terms, F23 over-enriched charged residues and under-enriched polar and aromatic residues.

**Residue level probability distribution of designed sequences and perplexity**. Natural protein sequences are constrained by evolution and explore only a part of the possible sequence space. *De novo* design methods, on the other hand, may explore all possible combinations guided by the energy functions to generate new proteins. With sequence recovery and divergence measurements, we have explored similarities between designed and native sequences. However, the question arises, for a given native residue, what is the range of residues the model considers plausible? Does the designed residue imitate the physicochemical

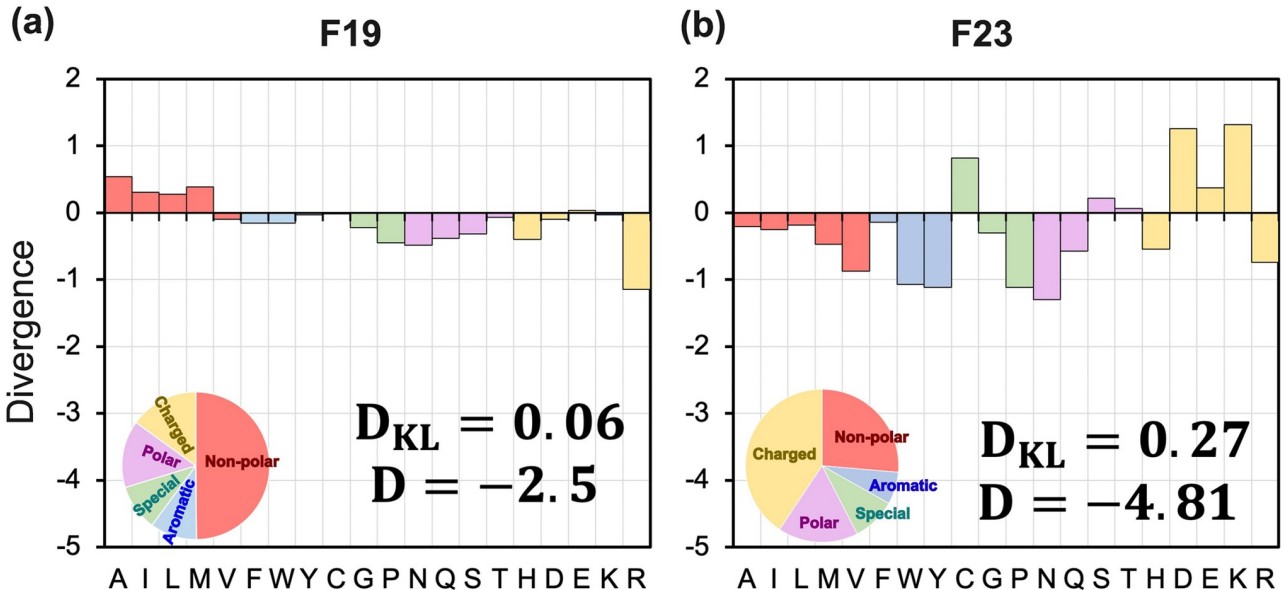

**Fig 10. Distribution of residue divergence in the redesigned membrane proteins.** Plots show the difference in log-likelihood ($D$) and KL-divergence ($D_{KL}$) of designed residue distribution relative to the native sequences. 204 MPs are redesigned with fix-bb and membrane orientation using the (a) F19 and (b) F23 energy functions. A positive divergence indicates that an amino acid is over-enriched, whereas a negative indicates that an amino acid is under-enriched. Values are given on a logarithmic scale. An amino acid composition pie chart for the sequence designed by each energy function is also shown in the bottom left-hand corner of the divergence plots. The color red is for non-polar, blue is for aromatic, violet is for polar, yellow is for charge, and green is for special residues.

properties of the native residue? And how confident is the model in its design prediction? To do so, we explored the probability distribution of designed amino acids at all positions with a given native residue (Fig 11). To measure the model's confidence in the prediction, we calculated perplexity per residue type (Fig 11d–11f):

$$pp_r = 2^{\sum_{s \in \text{AA}_{20}} -p(s|r)\log_2 p(s|r)} \tag{11}$$

where, $pp_r$ is the perplexity for a native amino acid type $r$, $p(s|r)$ is the probability of amino acid $s$ being designed to replace a native residue $r$. The probability $p(s|r)$ is given as:

$$p(s|r) = \frac{\sum_{i=1}^{N_{bb}} \sum_{j=1}^{N_i^{res}} \mathbb{1}_{x_{i,j}}^{des}(s, r)}{\sum_{i=1}^{N_{bb}} \sum_{j=1}^{N_i^{res}} \mathbb{1}_{x_{i,j}}^{nat}(r)} \tag{12}$$

where the superscripts des and nat denote designed and native sequences. The indicator function in the numerator counts native residues $r$ that convert to $s$ : $\mathbb{1}_{x_{i,j}}^{des}(s, r) := 1$ if $(x_{i,j}^{des} = s \wedge x_{i,j}^{nat} = r)$ and 0 otherwise, and the denominator indicator function counts native residues of type $r$ : $\mathbb{1}_{x_{i,j}}^{nat}(r) := 1$ if $x_{i,j}^{nat} = r$ and 0 otherwise. When a model assigns a high probability to a given residue, imply its confidence in the decision (meaning less perplexed). To elaborate, a perplexity of 20 means the model is confused with 20 residue choices for design, and a perplexity of 2 means the model is confused with 2 residue choices. However, low perplexity alone can be misleading, and it is important for the designed residue choices to be of the right kind. Thus, an ideal energy function will have a confusion matrix of residue types on design outcomes with a high probability for diagonal elements (near diagonal elements represent the native residue or of similar chemical properties) and low perplexity.

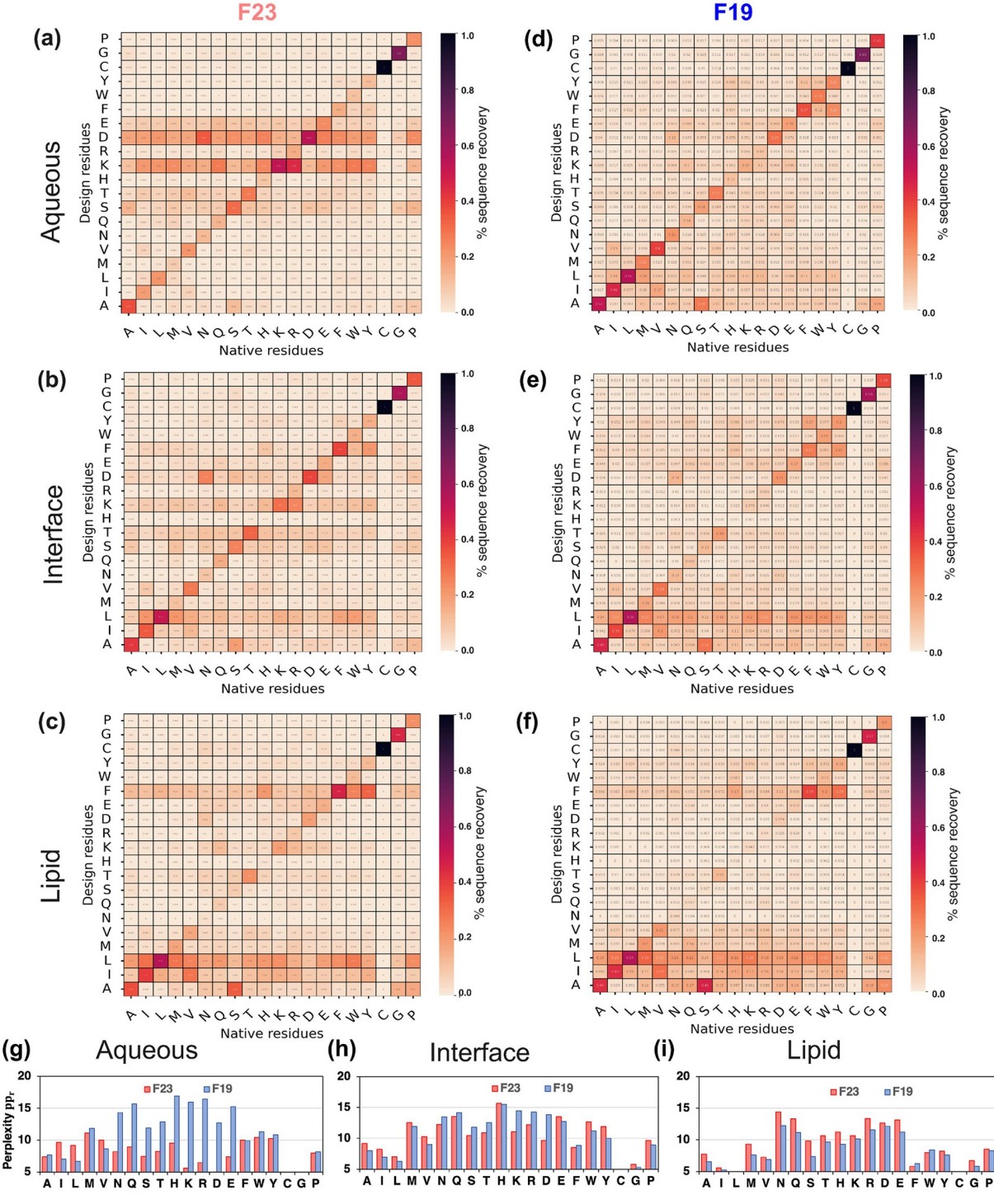

**Fig 11. Redesigned membrane proteins exhibit native-like sequences.** (a-c) Confusion matrices presenting probability distribution ($p_{i|r}$) of design residues $i$ replacing a native residue $r$ by F23 energy function. (d-f) Confusion matrices presenting probability distribution ($p_{i|r}$) of design residues $i$ replacing a native residue $r$ by F19 energy function. (g-i) Per-residue perplexity ($pp_r$) by F19 and F23 energy functions. In the confusion matrices, black denotes the highest (1.0), and light peach denotes the lowest (0.0) probabilities respectively. The criteria to distinguish between the membrane region is based on relative hydration in the membrane ($f_{hyd}$) and geometry of the water-exposed pore of the protein ($f_{pore}$). [44] Lipid: $f_{hyd} < 0.25$, $f_{pore} < 0.50$; Interface: $f_{hyd} \in [0.25, 0.75]$, $f_{pore} < 0.50$; Aqueous: $f_{hyd} > 0.75$, $f_{pore} > 0.50$.

F19 has low perplexity for designing non-polar and special residues and higher perplexity for charged and polar residues (Fig 11g–11i). F23 follows the trends of F19; however, it improved (reduced) the perplexity of the charged residues significantly for the aqueous region. For other polar residues, F23 has low perplexity; however, it often selects charged residues in place of other polar residues.

To sum, F19 can capture native protein-like features slightly better than F23. The perplexity and confusion matrix of F23 shows higher confidence and accuracy for designing charged residues.

## Conclusion

We developed a new implicit energy function Franklin2023 (F23) by adding electrostatic energy terms to the existing energy function Franklin2019 (F19) and evaluated its performance based on five different tests calculating structure, stability, and design calculations. F23 has three modifications relative to F19. First is the fast-to-compute Coulomb-based electrostatic energy, including the low dielectric constant of the lipid bilayer. Second is the mean-field-based electrostatic energy term capturing the effect of the lipid head group in the membrane environment. Third is the modified value of $\Delta G_{w,l}^{aa}$ for Asp and Glu residues at neutral pH. As a result, F23 captures electrostatic and non-polar properties of diverse homogeneous lipid layers of variable thickness, distinguishes between zwitterionic and anionic lipid head groups, and identifies pores in MP structures.

Fig 12 summarizes the performance of F19 and F23 on the diverse benchmark test set. Relative to F19, F23 has improved the calculation of the tilt angle of membrane proteins for 90% of WALP peptides of different lengths, 15% of TM-peptides, and 25% of the adsorbed peptides. To calculate the tilt angles, we used a fixed backbone for the peptides. Experimental results have shown that the peptide backbone is quite flexible; we hypothesize allowing backbone flexibility will further improve the tilt angle calculation. F19 and F23 predicted stability equally well in $\Delta\Delta G_{mutation}$ and $\Delta\Delta G_{ins}$. F19 identifies native protein-like residues better than F23.

| Test | Description | Experiment error | Computational metric | F19 | F23 |
|------|-------------|------------------|----------------------|-----|-----|
| **Protein orientation in the bilayer** | | | | | |
| 1a | Polyalanines tilt angle | — | | 100 | 100 |
| 1b | WALP peptides tilt angle | ±5° | %peptides with error | 60 | 80 |
| 2a | TM-peptide tilt angle | ±5° | ≤ ±20°↑ | 85 | 100 |
| 2b | Adsorbed peptide tilt angle | ±12° | | 60 | 85 |
| **Stability** | | | | | |
| 3a | $\Delta\Delta G_{mutation}$ (OmpLA) | | RMSE↓ | 6.40 | 4.88 |
| 3b | $\Delta\Delta G_{mutation}$ (PagP) | | RMSE↓ | 7.09 | 7.10 |
| 4 | $\Delta\Delta G_{w,l}$ for insertion | | $R^2$ ↑ | 0.98 | 0.99 |
| **Design** | | | | | |
| 5 | Sequence Recovery | | $D_{KL}$↓ | -2.5 | -4.8 |
| 5 | Sequence Recovery | | Recovery↑ | 0.31 | 0.30 |

**Fig 12. Summary of F19 and F23 performance on MP benchmark tests.** Darker boxes show better performances.

However, F23 significantly improved the confidence and accuracy of designing charged residues. For simplicity and lack of exact lipid types, we performed all design calculations in the DLPC bilayer; this is a broadly simplifying assumption that can be addressed by using the protein-matched membrane types.

The results of the benchmark tests show that, relative to F19, F23 has improved the ability of implicit membrane models to capture more relevant features of MPs. However, F23 did not improve predictions of stability and design performance as expected, possibly due to an incomplete representation of the membrane environment in the model. There are now several future steps that can be taken to improve the implicit models. Some potential steps are adding the effect of pH to account for membrane-induced pKa shifts, [68, 69] using robust optimization techniques to ameliorate double counting of physical effects due to use of empirical functions, [70] adding mixed and asymmetric lipid layers, [71, 72] and adding deformation of the lipid interface as it is perturbed by charged amino acids [73, 74]. F23 can serve as a good foundation model for including pH or lipid mixtures, as it includes additional electrostatic features and more characteristics of the lipid features. To capture the interface deformation or interaction of lipids with the residues, we have to move beyond the slab-like rigid representation of the membrane. Approaches such as those that include a defined curvature [75] or elastic bending constants of lipid membranes [76] along with other robust models can serve as foundation models for this purpose. We hypothesize that adding the above-mentioned features will capture the characteristics of realistic membranes. Configurations generated by implicit models can be used as an input or a filter for further investigations, including all-atom models to study phenomena mediated by lipid and water interaction. With continued improvement in MP feature representation, we anticipate that the implicit membrane model will enable reliable, high-throughput, and high-resolution MP structure prediction and design.

## Supporting information

**S1 Text. Implicit model to capture electrostatic features of membrane environment. Fig A. WALP35 peptide at the center of the lipid membrane**. (a) The total energy by F23 and $\Delta G_{w,l}^{aa}$ using F19 and F23 calculated as a function of tilt angles. (b) The net energy components due to the presence of the lipid membrane as a function of tilt angle calculated using F19(including $\Delta G_{w,l}$), F23 (including $\Delta G_{w,l}^{aa}$, $\Delta G_{lipid}^{aa}$ and M12 (including hydrophobicity, solvation, and knowledge terms). **WALP39 peptide in the lipid membrane**. (c) The per-residue $\Delta G_{w,l}^{aa}$ calculated using F23 and F19 for WALP39 at a depth = 10Å, tilt-angle = 90˚ and azimuthal-angle = 0˚. (d) The total energy by F23 and $\Delta G_{w,l}^{aa}$ using F19 and F23 for WALP39 calculated as a function of tilt angles. (e) The net energy components due to the presence of the lipid membrane at depth = 0Å and as a function of tilt angle calculated using F19, F23 and M12. The arrows show the corresponding axes for the plot. The ones for which arrows are not shown have values represented in the other direction. **Fig B. Comparing the energy landscape for WALP25 and WALP35 by M12 at the center of the lipid membrane**. The M12 total energy and other membrane-based energy terms were calculated for WALP25 and WALP35 peptides as a function of tilt angle at a depth = 0Å and minimized over rotation angle. The different score terms are as follows: (a) The total energy due to the membrane environment includes hydrophobicity, solvation, and knowledge terms. (b) the knowledge term is a statistical measure of the propensity of a residue to be at a particular membrane depth. (c) hydrophobicity energy and (d) solvation energy. **Fig C. Energy landscape of influenza A M2 peptide (pdb: 1mp6)**. (a) Comparing the native tilt angle (gray) with that calculated by F19 (blue) and F23(magenta and light pink). The Asp and c-term Leu for which the $\Delta G_{w,l}^{aa}$ is modified by F23 are shown in sticks. The total energy landscape of 1mp6 is shown as calculated by (b) F19 and (c) F23 as a function of depth

in the membrane and tilt angle. The energy at each tilt angle is minimized over all azimuthal angles. (d) The per residue $\Delta G_{w,l}^{aa}$ of 1mp6 calculated using F19 and F23 at membrane depth $d = 10\text{Å}$, tilt-angle = 0° and azimuthal-angle = 0°. The trend of (e) total energy, (f) $\Delta G_{w,l}^{\text{aa}}$, and (g) $\Delta G_{\text{lipid}}^{\text{aa}}$ as a function of tilt angles as calculated by F19 at membrane depth $d = 5\text{Å}$ and F23 at membrane depths $d = 5$ and $9\text{Å}$. **Fig D. Energy landscape of magainin-cecropin hybrid peptide (pdb: 1f0d)**. (a) Comparing the native tilt angle (gray) with that calculated by F19 (blue) and F23(magenta). The total energy landscape of 1f0d is shown as calculated by (b) F19 and (c) F23 as a function of depth in the membrane and tilt angle. The energy at each tilt angle is minimized over all azimuthal angles. The trend of (d) total energy, (e) $\Delta G_{w,l}^{\text{aa}}$, and (f) $\Delta G_{\text{lipid}}^{\text{aa}}$ as a function of membrane depth as calculated by F19 and F23 minimized over all tilt-angle and azimuthal-angle. **Fig E. Test1:Orientation of WALP in different lipid layers**. Predicted tilt angle of WALP peptides as a function of the number of residues. The WALP peptides are represented as WALPx, where x is the number of residues. The different markers indicate tilt angles calculated using F23 and different lipid types. The experimental tilt angles are measured in the DMPC lipid types. Filled markers present peptides for which experimental results are available, and those for which simulated data is available are shown by unfilled markers. **Fig F. Test6: Variation in the Rosetta designs for a given PDB backbone**. (a) Comparing the sequence recovery for a PDB backbone. The bar shows the mean and the error bar shows the variation in the sequence recovery of 50 Rosetta designs for a given PDB. (b) Comparing the total Rosetta score (REU) for a PDB backbone. The bar shows the mean and the error bar shows the variation in the total score of 50 Rosetta designs for a given PDB. **Table A**. Electrostatic potential parameters fit from all-atom molecular dynamics data. **Table B**. Equation to calculate the $\Delta G_{w,l}^{\text{atom}}$ transfer energy. **Table C**. Lipid composition parameters for $\alpha$-helical peptide tilt-angle calculations.
(PDF)

## Acknowledgments

We appreciate Dr. Rebecca Alford for sharing some of the code for the dielectric constant part of the work. We acknowledge Dr. Hope Woods and Dr. Rocco Moretti for reviewing our codes. We thank Prof. Damien Thevenin, Prof. Alexey Ladokhin for insightful discussions regarding WALP peptides. We also thank Dr. Patrick Fleming, Prof. Jeff Klauda, and Yulan Yu for their insightful discussions regarding electrostatic field verification. This work was carried out using facilities at the Advanced Research Computing at Hopkins (ARCH).

## Author Contributions

**Conceptualization:** Rituparna Samanta.

**Data curation:** Rituparna Samanta.

**Formal analysis:** Rituparna Samanta.

**Funding acquisition:** Jeffrey J. Gray.

**Investigation:** Rituparna Samanta.

**Methodology:** Rituparna Samanta.

**Project administration:** Jeffrey J. Gray.

**Software:** Rituparna Samanta.

**Supervision:** Jeffrey J. Gray.

**Validation:** Rituparna Samanta.

**Visualization:** Rituparna Samanta.

**Writing – original draft:** Rituparna Samanta.

**Writing – review & editing:** Jeffrey J. Gray.

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
