## [Decision Letter · Decision Letter 0]

25 Sep 2023

Dear Prof. Gray,

Thank you very much for submitting your manuscript "Implicit model to capture electrostatic features of membrane environment." for consideration at PLOS Computational Biology. As with all papers reviewed by the journal, your manuscript was reviewed by members of the editorial board and by several independent reviewers. The reviewers appreciated the attention to an important topic. Based on the reviews, we are likely to accept this manuscript for publication, providing that you modify the manuscript according to the review recommendations.

Here is an additional comment for your consideration based on my own reading (Nir Ben-Tal):

The literature survey includes recent publications. But continuous solvent models as well as coarse-grained models of peptide-membrane interactions date way back, into the previous millennium. The manuscript completely neglects to refer to these. For example, Turkan Haliloglu and myself have published a lot on this. And Skolnick, Efremov, Biggin, Sansom,... I'm giving here two references of my own, just because they are easy for me to find, but they refer to many relevant works also by others. Plenty to read... The authors may want to refer to some of the old works that outlines ideas that are now implemented more accurately. But please feel free to completely ignore this comment. Maybe I'm biased.

1) https://www.sciencedirect.com/science/article/abs/pii/S106358230252010X

2) https://www.ncbi.nlm.nih.gov/pmc/articles/PMC3394254/ 

End of comment.

Sincerely,

Turkan Haliloglu

Academic Editor

PLOS Computational Biology

Nir Ben-Tal

Section Editor

PLOS Computational Biology

Reviewer's Responses to Questions

**Comments to the Authors:**

Reviewer #1: In this work, the authors developed an implicit membrane model (F23) by implementing modifications to a prior model (F19). Modifications include a mean field electrostatic potential based on the atomistic simulations, position dependent dielectric constant term along the membrane normal and a modified energy term for the water to bilayer partitioning for Asp and Glu. Overall, they obtained a moderate improvement over the F19 model, especially for the orientation of membrane peptides/proteins. The most significant addition of the model is to develop the mean field potential using different lipid types, which can allow to study proteins in lipid models with various head groups, saturation level, hydrocarbon chain length. This paper is a valuable contribution to membrane model development and will be of interest for PLOS Computational readers. Below, I listed my comments for improving the manuscript, mostly about providing more computational details to ensure reproducibility.

1- Figure 1 shows the electrostatic potentials based on the atomistic simulations. However, no details were provided about these simulations. I understand that they came from a prior work, an explanation for confirming this would be good. But, otherwise, if the atomistic simulations are part of this work, simulation details should be provided (system size/preparation, simulation time, ensemble, force field, integrator etc.).

2- For the tilt angle analysis (Figure 4), it is not clear how the lowest energy orientations of peptides were obtained (lines 204-205). Were they obtained by a Monte Carlo sampling, or were any MD simulations performed? Was a previously established protocol used, or they have developed a new protocol? More details about how these orientations were predicted would be good for reproducibility of the data.

3- Also, which initial structures were used for both AAx and WALP peptides?

4- For predicting orientation of the peptides (AAx and WALP, TM and adsorbed), which electrostatic potential was used for F23, from which lipid model, what length, and head group? Authors mentioned using DLPC for WALP peptides (lines 221-223), but it is not clear which lipid type was used for other peptides/proteins. Peptide orientations will vary significantly according to the membrane width, so the membranes (lipid type) used in the experiments/simulations and in their models should be provided (maybe in a table) for making a meaningful comparison.

5- The authors mentioned that the model is fast-to-compute. It would be good to add some benchmarking about the computing time performances.

Reviewer #2: Summary:

The authors present updates to the Rosetta membrane score function that include two new score terms, an electrostatic energy term and a membrane dependent dielectric constant, and a modification to an existing membrane dependent score term. They show results from multiple tests for the previous score functions and the new score function. The paper is well written and organized. The improvements to the Rosetta membrane score function adds an important contribution to computational membrane protein structure prediction. Some of the choices the authors made could be justified or explained in more detail. It is unclear from the data presented how variable the results for different tests are when comparing independent runs. The work could benefit from a more detailed supplementary method section or protocol captures.

Major Comments:

• The authors should provide more details on how weights were optimized for the three membrane dependent energy terms. The authors say the weights were optimized to maximize the correlation coefficient for experimentally measured ddGs and then cite two papers. One paper is the hydrophobicity scale from Moon and Fleming and the other is a review on the Poisson-Boltzmann equation. Is that the correct reference, reference 35 on line 171 in the “Integration of new terms into F23” section? What protocol was used to calculate the predicted ddGs with F23 that were used for the weight optimization? Are there any concerns with fitting the weights using the same experimental data that the transfer energy from water to bilayer term is based on? How/why were Gly, Ala, Pro, Asp, and Glu chosen to be excluded for overfitting? Is there any way to optimize these weights using more than one task?

• How much variation is there in ddG predictions? Could multiple runs be done with mean and standard deviation be shown in scatter plots?

• How consistent are design results? Were these ran multiple times or just once?

Minor Comments:

• F19 has 13 different lipid options. It is unclear if F23 has the same lipid options or only some of them

• Figure 3B: Were the large shifts in dGw,l for COO and OOC in F23 compared to F19 expected? Is there any significance that OOC for F23 is now almost the same as in IMM1?

• In the introduction the authors mention a previously developed 12-test benchmark. Could the authors either show results, even just as part of the supplementary materials, for all the 12 test or provide rational behind the test they chose to show? For example, there no docking results shown.

• What protocol is used to determine the tilt angle for peptides?

• I find it odd that in the polyalanine tilt angle test M12 does not recapitulate the results from the 5-slab model, but the argument for why M12 matches the 5-slab model for the WALP peptides is they are based on the same hydrophobicity scale.

• Were different lipid compositions used for Test 2? How much does the lipid composition effect the results of tilt angle predictions?

• Fix quality of figure S2 so the faint outline around the plots are not seen

• What is the little blue dot in Figure 7A above the F23 datapoint for Lysine?

• In Figure 7, could the authors include which color line and text correspond to which energy function.

• In Figure 9, datapoints are shown for energy functions not previously introduced.

Reviewer #3: In this article, the authors have developed a new implicit energy function, Franklin2023 (F23), as an extension of the Franklin2019 (F19) energy function. In the previous article [ref. 37], the authors introduced a 12-test benchmark for the energy function, which was tested on F2019 and identified areas for potential improvement. In this article, several of these suggestions have been implemented, particularly focused on improvements in the representation of membrane electrostatics and the correction of water-to-lipid transfer energies for Glu and Asp residues. The F23 function was validated through several tests, and the results were compared with F2019 and other energy functions from the literature.

It is evident that a significant disparity exists between our current understanding of the structures (and function) of soluble and membrane-related proteins. This underscores the imperative to advance methods for studying proteins within lipid-rich environments. A promising approach lies in implicit energy functions that enable fast calculations of free energies, potentials and experimentally measurable quantities such as tilt angles, and are also very valuable tools in sequence design. Therefore, the topic and methodology presented in this article are very relevant and of significant interest both to the readers of PLOS Computational Biology and to the wider scientific community.

The article is well-written, with effective presentation of methods and results. It is, in my opinion, ready for publication, with only minor corrections, which are listed below, along with a few comments:

1. Please enhance the quality of some figures and add missing details to the captions for the benefit of readers. a) In Figures 4 and 5, some peptide names are partially missing or overlapped. b) It is unclear what the lines in Figure 4 represent. Please add descriptions to the caption. c) Figure 6 is referred to as Figure 6b in the text, and it would be useful to have a more detailed explanation of Figure 6 data in the caption. d) In Figure 9, specify what the empty and filled symbols represent. e) It would be useful to provide a clearer explanation of what the data in Figure 10 represents. f) Please add a more detailed description of Figure S1 data.

2. Please check if the numbering of tests in the final table is accurate.

3. The statement about the lower hydrophobicity of polar residues contributing to a more accurate prediction of the tilt angle and its confirmation in Fig. S1e is a bit puzzling to me. Could the authors rewrite this sentence to make it clearer.

4. Regarding the last test aimed at the protein design, it is not optimal that F23 performed worse than F19 (or nearly the same). However, the improvement in the percentage of sequence recovery for surface-exposed residues, in contrast to buried residues, is promising. Could this indicate a potential path for improvement, possibly by adjusting the representation of the dielectric constant of the bilayer?

5. It is interesting to note that the improvements in predicting stability and design performance were not observed with F2023 as probably expected. The addition of terms to enhance the electrostatic features of the membrane should ideally lead to improvements in all tests. This result may suggest that some essential aspects are still missing or that the included terms fail to represent certain critical contributions. Please could you comment on this in the article.

6. The authors have outlined future directions for energy function development, which involve adding terms to provide an even more detailed description of the features of the lipid environment, such as pH effects. Does this imply that the authors consider F23 as a good foundation upon which to build by adding new terms?

**Have the authors made all data and (if applicable) computational code underlying the findings in their manuscript fully available?**

Reviewer #1: **No: **The protocols to obtain orientations of the peptides, and the lipid type used for each peptide were not provided.

Reviewer #2: Yes

Reviewer #3: Yes

PLOS authors have the option to publish the peer review history of their article (what does this mean?). If published, this will include your full peer review and any attached files.

Reviewer #1: No

Reviewer #2: No

Reviewer #3: No

Figure Files:

Data Requirements:

Reproducibility:

References:

---

## [Decision Letter · Decision Letter 1]

13 Dec 2023

Dear Prof. Gray,

We are pleased to inform you that your manuscript 'Implicit model to capture electrostatic features of membrane environment.' has been provisionally accepted for publication in PLOS Computational Biology.

Best regards,

Turkan Haliloglu

Academic Editor

PLOS Computational Biology

Nir Ben-Tal

Section Editor

PLOS Computational Biology

Reviewer's Responses to Questions

**Comments to the Authors:**

Reviewer #1: The authors sufficiently addressed my comments and I recommend this manuscript for publication.

Reviewer #2: The authors have addressed all concerns. This work will be of great interest to the membrane protein structure community.

**Have the authors made all data and (if applicable) computational code underlying the findings in their manuscript fully available?**

Reviewer #1: Yes

Reviewer #2: Yes

PLOS authors have the option to publish the peer review history of their article (what does this mean?). If published, this will include your full peer review and any attached files.

Reviewer #1: No

Reviewer #2: No

---

## [Editor Report · Acceptance letter]

12 Jan 2024

PCOMPBIOL-D-23-00999R1 

Implicit model to capture electrostatic features of membrane environment.

Dear Dr Gray,

I am pleased to inform you that your manuscript has been formally accepted for publication in PLOS Computational Biology. Your manuscript is now with our production department and you will be notified of the publication date in due course.

With kind regards,

Zsofi Zombor
